# Tunnel field-effect transistors for sensitive terahertz detection

I. Gayduchenko[1,2,10], S. G. Xu [3,4,10], G. Alymov[2,10], M. Moskotin[1,2], I. Tretyakov[5], T. Taniguchi[6], K. Watanabe [7], G. Goltsman[1,8], A. K. Geim[3,4], G. Fedorov [1,2✉], D. Svintsov[2✉] & D. A. Bandurin [2,3,9✉]

The rectification of electromagnetic waves to direct currents is a crucial process for energy harvesting, beyond-5G wireless communications, ultra-fast science, and observational astronomy. As the radiation frequency is raised to the sub-terahertz (THz) domain, ac-to-dc conversion by conventional electronics becomes challenging and requires alternative rectification protocols. Here, we address this challenge by tunnel field-effect transistors made of bilayer graphene (BLG). Taking advantage of BLG's electrically tunable band structure, we create a lateral tunnel junction and couple it to an antenna exposed to THz radiation. The incoming radiation is then down-converted by the tunnel junction nonlinearity, resulting in high responsivity (>4 kV/W) and low-noise (0.2 pW/$\sqrt{\text{Hz}}$) detection. We demonstrate how switching from intraband Ohmic to interband tunneling regime can raise detectors' responsivity by few orders of magnitude, in agreement with the developed theory. Our work demonstrates a potential application of tunnel transistors for THz detection and reveals BLG as a promising platform therefor.

[1] Physics Department, Moscow Pedagogical State University, Moscow 119435, Russia. [2] Moscow Institute of Physics and Technology (National Research University), Dolgoprudny 141700, Russia. [3] School of Physics, University of Manchester, Oxford Road, Manchester M13 9PL, UK. [4] National Graphene Institute, University of Manchester, Manchester M13 9PL, UK. [5] Astro Space Center, Lebedev Physical Institute of the Russian Academy of Sciences, Moscow 117997, Russia. [6] International Center for Materials Nanoarchitectonics, National Institute of Material Science, Tsukuba 305-0044, Japan. [7] Research Center for Functional Materials, National Institute of Material Science, Tsukuba 305-0044, Japan. [8] National Research University Higher School of Economics, Moscow 101000, Russia. [9] Present address: Department of Physics, Massachusetts Institute of Technology, Cambridge, MA 02139, USA. [10] These authors contributed equally: I. Gayduchenko, S. G. Xu, G. Alymov. ✉email: gefedorov@mail.ru; svintcov.da@mipt.ru; bandurin@mit.edu

ield effect transistors (FETs) have recently found an unexpected application for the rectification of THz and sub-THz signals beyond their cutoff frequency[1,2]. This technology paves the way for on-chip[3], low-noise[4], and sub-nanosecond radiation detection[5,6] enabling ≳10 Gb/s data transfer rates. Contrary to competing diode rectifiers, FETs offer the possibility of phase-sensitive detection[7,8] vital for noise-immune communications with phase modulated signals. Furthermore, recent innovations towards enhanced responsivity of FET detectors include the use of two-dimensional (2D) materials[9–11], exotic nonlinearities[12–16], enhanced light–matter coupling[17] and plasmonic effects[18–20] bringing FET-enabled detection to the center of THz technology. Despite the rich and complex physics of THz rectification, the responsivity of most FET-detectors is governed by the sensitivity of the channel conductivity $G_{ch}$ to the gate voltage $V_g$, parameterized via the normalized transconductance $F = -\mathrm{d}\ln G_{ch}/\mathrm{d}V_g$[2,21]. The transconductance in conventional FETs has a fundamental limit of $e/k_BT$ ($e$ is the elementary charge and $k_BT$ is the thermal energy) dictated by the leakage of thermal carriers over the gate-induced barrier, termed as "Boltzmann tyranny". Although this process is well-recognized as a limiting factor for the minimal power dissipation of FETs in integrated circuits, it has been scarcely realized that it also imposes a bound on the responsivity of antenna-coupled FETs to THz fields.

One of the most promising routes to escape from the Boltzmann tyranny is the manipulation of interband tunnelling instead of intraband thermionic currents. This idea is materialized in a tunnel field effect transistor (TFET)[22–24]. TFETs find their applications in low-voltage electrical and optical switching[25], accelerometry[26], chemical[27] and biological sensing[28,29]. In spite of this variety, the use of TFETs for the rectification of high-frequency signals[30] has not been attempted so far. This is also surprising considering recent advances in the development of tunnelling high-frequency rectifiers and detectors based on quantum dots[31,32], diodes[33–38] and superconducting tunnel junctions[39–41]. A possible reason is that the low on-state current and relatively small cut-off frequency of TFETs have stimulated the belief on their inapplicability in teratronics[42].

In this work, we show that the opposite is true and demonstrate the use of TFETs for highly sensitive sub-THz and THz detection. Using bilayer graphene (BLG) as a convenient platform for this enquiry, we fabricate a dual-gated TFET and couple it to a broadband THz antenna. The received high-frequency signal is rectified by electrostatically defined tunnel junction resulting in high-responsivity and low-noise detection. Our experimental results and the developed theory suggest that the origin of the high responsivity in our detectors is not the large transcoductance, but rather steep curvature of the tunnelling $I-V$ characteristic[43]. Our findings point out that even TFETs without sub-$k_BT/e$ switching can act as efficient THz rectifiers preserving all the benefits of transistor-based detection technology.

## Results

**Device fabrication and characterization**. For the proof-of-principle demonstration, we constructed a TFET of a BLG taking advantage of its unique electronic properties. BLG is a narrow-band semiconductor characterized by a tunable band structure highly sensitive to the transverse electric field[44]. This ensures a steep ambipolar field effect and allows for an independent control of the band gap size and the carrier density, $n$[45], providing a unique opportunity for a fully electrostatic engineering of the spatial band profile[46–48]. We employed this property to electrostatically define typical TFET configuration shown in Fig. 1a, b. In addition, BLG hosts a high-mobility electronic system, a crucial property for high-frequency applications. As we now proceed to show, these properties make BLG a

convenient platform to demonstrate the drastic differences in performance of intraband field-effect-enabled detection and its interband tunnelling counterpart within the same device.

We fabricated our detector by an encapsulation of BLG between two slabs of hexagonal boron nitride (hBN) using standard dry transfer technique described elsewhere[49] (see "Methods"). The BLG channel of length $L = 2.8\,\mu$m and width $W = 6.2\,\mu$m was assembled on top of a relatively thin (~10 nm) graphite back gate which ensured efficient screening of remote charge impurities in Si/SiO$_2$ substrate[50]. The device was also equipped with a second (top) gate electrode deposited symmetrically between the source and drain contacts. Importantly, relatively short ($l < 100$ nm) regions near the contacts were not covered by the top gate and thus were affected by the bottom one only. This configuration allowed us to define a lateral tunnel junction between single- and double-gated regions when the top and bottom gate voltages ($V_{tg}$ and $V_{bg}$, respectively) had opposite polarities[46–48], as explained in Fig. 1b. The device was coupled to the incident radiation via a broadband bow-tie antenna connected to the source and top-gate electrodes. The rectified dc photo-voltage, $V_{ph}$, was read out between the source and drain terminals as shown in Fig. 1a (see "Methods").

Prior to photoresponse measurements, we characterized the transport properties of our device. Figure 1e shows the dependence of our detector's two-terminal resistance, $r_{2pt}$, on $V_{tg}$ for two representative values of $V_{bg}$ measured at $T = 10$ K. At $V_{bg} = 0$ V, $r_{2pt}(V_{tg})$ exhibits familiar bell-like structure that peaks at the charge neutrality point (CNP) where $r_{2pt} \approx 0.4$ kΩ (inset of Fig. 1e). Application of $V_{bg} = 2$ V shifts the CNP to negative $V_{tg}$ and results in drastic increase of $r_{2pt}$ that reaches 20 kΩ already at $V_{tg} \approx -3.5$ V. This increase is a clear indicative of the electrically induced band gap in BLG[44,45].

**Tunnelling-enabled detection**. Figure 2a shows the external responsivity of our detector, $R_v = V_{ph}/P_{in}$, as a function of $V_{tg}$ recorded in response to $f = 0.13$ THz radiation. Here $V_{ph}$ is the generated photovoltage and $P_{in}$ is the incident radiation power (see "Methods" for the details of responsivity determination). At $V_{bg} = 0$ V, $R_v(V_{tg})$ exhibits a standard antisymmetric sign-changing behaviour with $|R_v|$ reaching 200 V/W close to the CNP. The functional form of $R_v(V_{tg})$ follows that of the normalized transconductance $F = -(\mathrm{d}G_{ch}/\mathrm{d}V_{tg})/G_{ch}$ (Fig. 2b), where $G_{ch} = 1/r_{2pt}$, and is consistent with previous studies of graphene-based THz detectors[10,19,51]. This standard behaviour is routinely understood in terms of a combination of resistive self-mixing and photo-thermoelectric rectification, two predominant mechanisms that govern THz detection in graphene-based FETs[52].

The response of our device changes drastically when a finite vertical electric field is applied perpendicular to the BLG channel. Figure 2a shows the $R_v(V_{tg})$ dependence for $V_{bg} = 1.5$ V and reveals a giant increase of $R_v$ exceeding 3 kV/W (red curve). A notable feature of the observed dependence is its strong asymmetry with respect to zero $V_{bg}$ behaviour: namely, $|R_v|$ is more than an order of magnitude larger for the p-doped channel (to the left from the CNP in Fig. 1e) as compared to the case of n-doping (to the right from the CNP in Fig. 1e). In addition, while the response decays rapidly with increasing $V_{tg}$ on the n-doped side, a finite $R_v$ is observed over the whole span of $V_{tg}$ at which the channel is p-doped. Furthermore, when the sign of $V_{bg}$ is reversed, $R_v(V_{tg})$ remains asymmetric but, in this case, it is strongly enhanced for the case of n-doped channel (blue curve in Fig. 2a and the top inset of Fig. 2a). Importantly, the $F$-factor remains fairly symmetric for the $V_{tg}/V_{bg}$ combinations at which $R_v$ exhibits strong asymmetry. This observation suggests that the strong rectification of THz radiation in our device is not caused

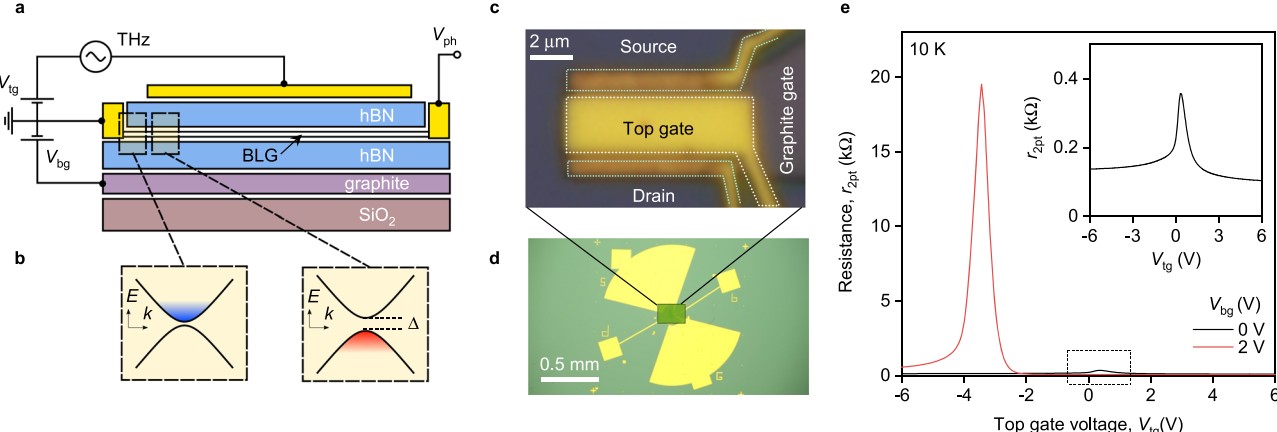

**Fig. 1 Dual-gated bilayer graphene THz detector. a** Schematic of an hBN encapsulated dual-gated BLG transistor. THz radiation is incident on a broadband antenna connected to the source (S) and gate terminals yielding modulation of the top gate-to-source voltage ($V_{tg}$) while the back gate voltage ($V_{bg}$) is fixed. The build-up photovoltage $V_{ph}$ is read out between the source and drain (D) terminals. **b** Band structure of the BLG at the interface between the n-doped bottom gate-sensitive region and dual-gated p-doped channel ($\Delta$ is the induced band gap). Blue and red colours illustrate conduction and valence bands fillings, respectively. Note, even for a single-gated region, a finite band gap appears in the energy dispersion due to the difference in on-site energies between the top and bottom graphene layers[44]. **c, d** Optical photographs of the fabricated dual-gated detector. The source and top-gate terminals are connected to a broadband bow-tie antenna. **e** The two-terminal resistance of our BLG device, $r_{2pt}$, as a function of $V_{tg}$ for two representative $V_{bg} = 0$ and $V_{bg} = 2$ V. Inset: Zoomed-in $r_{2pt}(V_{tg})$ for $V_{bg} = 0$ V. $T = 10$ K.

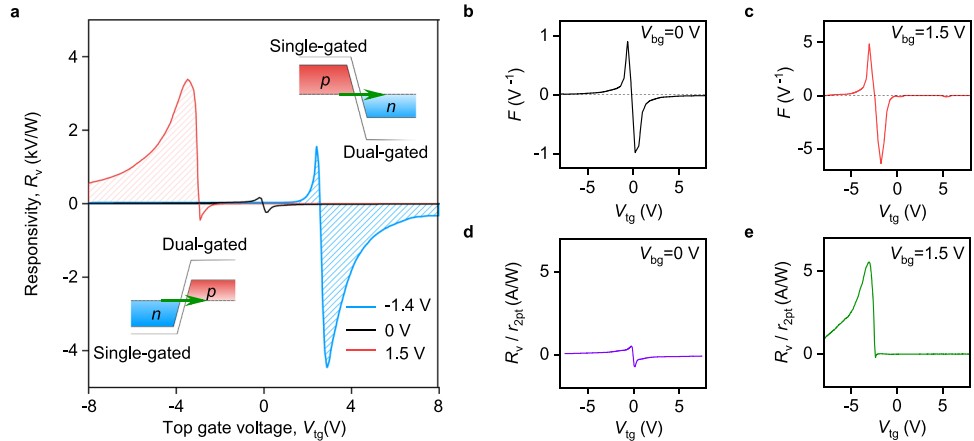

**Fig. 2 Tunnelling-assisted THz detection. a** Detector responsivity, $R_v$, as a function of $V_{tg}$ for $V_{bg} = 0$ V (black), $V_{bg} = -1.4$ V (blue) and $V_{bg} = 1.5$ V (red) measured in response to $f = 0.13$ THz radiation. $T = 10$ K. Inset illustrates band profiles in the vicinity of the single and dual-gated interface when $V_{bg}$ and $V_{tg}$ are of opposite polarities. Green arrows illustrate interband tunnelling. **b, c** Normalized transconductance $F$ versus $V_{tg}$ obtained by numerical differentiation of the device resistance for $V_{bg} = 0$ V (**b**) and $V_{bg} = 1.5$ V (**c**). Note, $F(V_{tg})$ dependencies are fairly symmetric whereas the $R_v(V_{tg})$ is highly asymmetric for the same $V_{bg}$ (**a**). **d, e** $R_v$ from **a** normalized to the channel resistance $r_{2pt}$ as a function of $V_{tg}$ for given $V_{bg}$.

by the nonlinearities in the dual-gated BLG channel. Moreover, the increase in $R_v$ cannot be explained by a trivial enhancement of channel resistance. To demonstrate this, in Fig. 2d, e we plot $R_v$ normalized to $r_{2pt}$, a quantity with the dimension of current responsivity. A symmetric $R_v/r_{2pt}(V_{tg})$ dependence measured at $V_{bg} = 0$ V is conceded with an amplified and highly asymmetric curve at finite $V_{bg}$, thereby excluding resistance-enabled $R_v$ enhancement.

Figure 3a, b detail our observations further by showing maps of $R_v(V_{tg}, V_{bg})$ and $r_{2pt}(V_{tg}, V_{bg})$. Enhanced $R_v$ is observed in two distinct quadrants characterized by an antisymmetric (with respect to the $V_{bg}$) sign pattern (see Supplementary Note 1 for the line cuts of the map in Fig. 3a). Outside these quadrants, $R_v$ was found negligibly small. Interestingly, $r_{2pt}$-map is fairly symmetric featuring gradual increase of resistance at the CNP with increasing vertical field as expected for BLG[44,45]. We have also studied the performance of our detectors at higher $f$ and

found consistent highly asymmetric response similar to that shown in Fig. 2b (Supplementary Note 2) highlighting broadband character of the rectification mechanism. Furthermore, using Johnson–Nyquist relation for the noise spectral density $S = \sqrt{4 k_B T r_{2pt}}$, we estimate the noise-equivalent power of our detector, NEP $= S/R_v$, to reach 0.2 pW$/\sqrt{\text{Hz}}$ at $T \approx 10$ K (Fig. 3c). For comparison, commercial superconducting hot electron bolometers (SHEB) operating at lower $T = 4.2$ K feature NEP of 0.1–2 pW$/\sqrt{\text{Hz}}$ (Fig. 3c, green shaded area) that makes our dual-gated detectors competitive with the commercial technology (Supplementary Note 3).

In order to understand deeper the peculiar detection mechanism of our cooled detector, we have studied the temperature dependence of its performance. Figure 3d compares the $R_v(V_{tg})$ dependencies measured at $T = 10$ and 77 K in response to $f = 0.13$ THz radiation. For p-doped channel, $R_v(V_{tg})$ drops by more

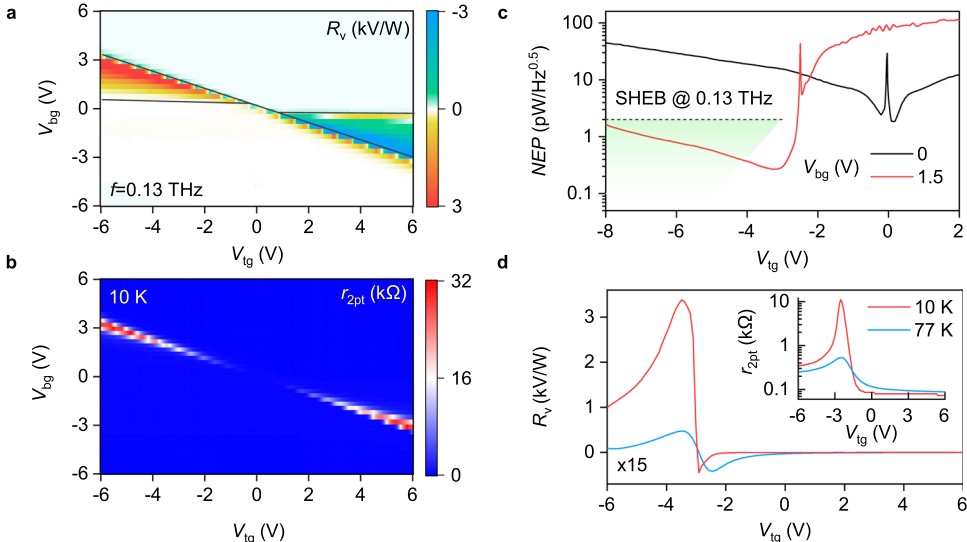

**Fig. 3 Performance of the BLG TFET detector. a** Responsivity of our detector as a function of $V_{bg}$ and $V_{tg}$ recorded in response to $f = 0.13$ THz radiation. The black lines demark the ($V_{tg}$, $V_{tg}$) regions where the tunnel junction configuration is realized. **b** $r_{2pt}(V_{tg}, V_{tg})$ map measured at $T = 10$ K. The appearance of highly resistive regions (red) points out to the band gap opening in BLG. **c** NEP of our detector at given $V_{bg}$ determined using the Johnson–Nyquist relation for the noise spectral density. Horizontal line marks NEP level for SHEBs operating at the same $f$ and $T = 4.2$ K (see Supplementary Note 3 for a detailed comparison of the BLG-TFET with other THz detectors). Green shaded region indicates the spread in NEP for SHEBs at higher $f$. **d** Temperature dependence of $R_v(V_{tg})$ and $r_{2pt}(V_{tg})$ (inset) at $V_{bg} = 1.5$ V.

than two orders of magnitude whereas a 10-fold decrease in $R_v(V_{tg})$ is observed for the n-doped side. Furthermore, in contrast to the behaviour observed at 10 K, the $R_v(V_{tg})$ curves become more symmetric at liquid nitrogen $T$. To compare, $r_{2pt}$ at the CNP also drops with increasing $T$ (inset of Fig. 3d), demonstrating usual insulating behaviour of gapped BLG at zero doping. However, $r_{2pt}$ exhibits clear asymmetry with respect to the CNP. In particular, for the case of n-doped channel, $r_{2pt}$ is rather small ($\approx 100 \, \Omega$) and it grows with increasing $T$, a signature of phonon-limited transport, whereas on the p-doped side we observed a pronounced decrease of $r_{2pt}$ with increasing $T$; $r_{2pt}$ is of the order of $0.5 \, k\Omega$ away from the CNP. The giant enhancement of $R_v$, insulating $T$-dependence of $r_{2pt}$ and its increase, when $V_{tg}$ and $V_{bg}$ are of opposite polarities, suggest that the behaviour of our BLG detector is governed by the interband tunnelling as we now proceed to demonstrate.

**Modelling tunnelling-enabled photoresponse**. Our dual-gated BLG transistor can be modelled by an equivalent circuit described in "Methods" (see below). It consists of the gate-controlled channel conductance $\tilde{G}_{ch}$ and tunnel junctions at the source and drain with conductances $G_S$ and $G_D$, respectively. The net responsivity $R_v$ of such a circuit is the sum of three "intrinsic" responsivities (marked with subscript $i$) weighted with voltage division factors $\gamma = [1 + G_S/\tilde{G}_{ch}]^{-1}$ and multiplied by a factor of $\approx 4Z_{rad}$ relating the mean square of the antenna's output voltage with the incident power ($Z_{rad}$ is the radiation resistance of the antenna; exact expression for the prefactor is given in "Methods"):

$$R_v \approx 4Z_{rad}[R_{TJ,i}|\gamma|^2 + R_{TG,i}\text{Re}\,\gamma + R_{ch,i}|1 - \gamma|^2]. \quad (1)$$

The channel responsivity, $R_{ch,i}$, is proportional to the transconductance[21] and appears due to resistive self-mixing effect, i.e. due to simultaneous modulation of carrier density by transverse gate field and their drag by longitudinal field. The tunnel junction responsivity $R_{TJ,i}$ emerges due to nonlinear dependence of tunnelling current on junction voltage, $V_{TJ}$. Finally, the responsivity

$R_{TG,i}$ appears due to the simultaneous action of the gate voltage that modulates tunnel barrier and junction voltage that pulls the carriers. All three contributions can be calculated from the sensitivities of conductances $G_S$ and $\tilde{G}_{ch}$ to $V_{tg}$ and $V_{TJ}$ ("Methods" and Supplementary Note 4).

Figure 4a plots the results of such calculations in a form of 2D map which shows $R_v$ dependence on $V_{tg}$ and $V_{bg}$ (see "Methods" and Supplementary Note 4). The map captures well all the features of the experiment, in particular, the asymmetric gate voltage dependence of the responsivity and its giant increase when the voltage of top and bottom gates is of the opposite polarity. This is most clearly visible in Fig. 4c which compares $R_v(V_{tg})$ dependencies for the cases of zero and finite $V_{bg}$. Moreover, our model indicates a broadband character of the tunnelling-assisted photoresponse (Supplementary Note 2) as well as provides a remarkable quantitative agreement with experiment provided that $Z_{rad} \approx 75 \, \Omega$, a typical value for the antenna of this type[19,52].

The peculiar response of our detector can be understood with the band diagrams shown in Fig. 4b. The detector can operate in two regimes: the regime of intraband transport (white, grey, yellow, and purple symbols on the map in Fig. 4a) and the regime of interband tunnelling (green and blue symbols on the map in Fig. 4a), depending on the gate voltage configuration. At zero $V_{bg}$, BLG is practically gapless, so that the tunnel barrier between the source and the channel is almost absent (white and grey symbols on the map in Fig. 4a). In this regime, the device responsivity is controlled by $R_{ch,i}$ which exhibits a symmetric dependence on $V_{tg}$ (cf Fig. 4c (black line) and Fig. 2a). On the contrary, when a finite bias is applied to the bottom gate, the tunnel junction is formed as illustrated in Fig. 4b (green and blue symbols). Its intrinsic rectifying capability $R_{TJ,i}$ exceeds that of the transistor channel $R_{ch,i}$, as shown in the inset to Fig. 4c by several orders of magnitude. This stems from an ultra-strong, exponential sensitivity of the tunnel conductance to the voltage at the junction, as compared to the smooth dependence of $\tilde{G}_{ch}$ on $V_{tg}$. Moreover, the ac voltage being rectified drops almost completely on a weakly conducting junction but not on the well-conducting

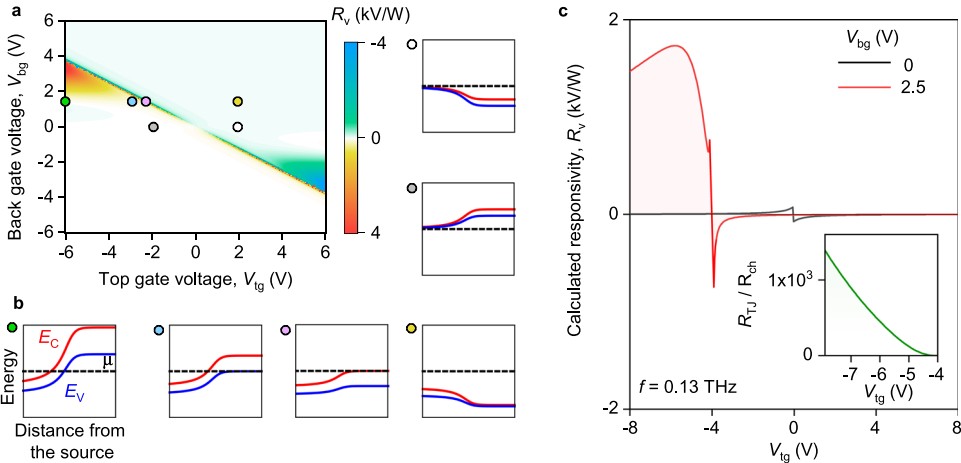

**Fig. 4 Modelling tunnelling-assisted THz detection. a** Calculated $R_v(V_{tg}, V_{bg})$ map of our dual-gated BLG device in response to $f = 0.13$ THz radiation. **b** Calculated band profiles for different $(V_{tg}, V_{bg})$ configurations indicated by the coloured symbols in **a**. White, grey, yellow, and pink symbols point to the band diagrams of the FET mode whereas the green and blue symbols correspond to the regime of interband tunnelling. Red, blue, and black lines illustrate conduction band minimum ($E_C$), valence band maximum ($E_V$), and the chemical potential ($\mu$), respectively. **c** Line cuts of the map in (a) for $V_{bg} = 0$ V and $V_{bg} = 2.5$ V. The radiation resistance of the antenna $Z_{rad} \approx 75 \, \Omega$ was used for these calculations ("Methods"). Inset: The ratio between the tunnel junction, $R_{TJ}$, contribution to the responsivity and that of the channel nonlinearity, $R_{ch}$, for $V_{bg} = 2.5$ V.

channel in the tunnelling regime ($|\gamma| \rightarrow 1$). This can be viewed as the "self-localization" of the ac field in the tunnelling rectifier, which contributes to the responsivity enhancement.

Our theory, which successfully describes the response of the BLG device, can also serve to demonstrate the prospects and fundamental limits of TFET-based THz detectors. In the current device, the tunnelling is assisted by fluctuations of in-plane electric field induced by local groups of charged impurities[53]. In ideal devices, the responsivity would exceed 100 kV/W, according to the model calculations (Supplementary Note 5). It is also remarkable that the expected high transconductance of TFET concedes to even higher nonlinearity of the tunnel junction, thus $R_{TJ,i} \gg R_{TG,i}$ in the present and ideal devices. $R_{TG,i}$ can dominate in situations where the source and channel simultaneously possess large gap and remain undoped; thence electron tunnelling occurs from a filled valence band of the source to an empty conduction band of the channel in the vicinity of band edges. Large density of states near the bottom of "Mexican hat"-like spectrum of BLG further increases TFET switching steepness[54]. Realization of such band alignment is possible with the application of the drain bias and/or with extra doping gates.

Last, we point to an important advantage of TFET rectifiers with 2D channels, namely, the low internal capacitance of the lateral tunnel junctions. Up to small logarithmic terms, it is given by[55] $C_{TJ} = 2W\varepsilon\varepsilon_0 \approx 0.4$ fF for the device width of $W = 6.2$ μm (as in our experiment) and dielectric environment $\varepsilon \approx 4$. The detection cutoff associated with the capacitive shunting of the tunnel junction is therefore expected at $f \sim 1/(2\pi C_{TJ}Z_{rad}) \sim 5$ THz for $Z_{rad} \approx 75 \, \Omega$.

## Discussion
Despite the fact that our model successfully describes all the features of the observed photoresponse, it does not account for a possible thermoelectric contribution to the responsivity of TFET detectors[5,10]. Assuming that the Seebeck coefficient varies between $S_{cont}$ in the single-gated region near the source contact and $S_{ch}$ in the double-gated channel, one can estimate the thermoelectric contribution as[52] $R_{TE} \approx (3e/2\pi^2 k_B)(S_{cont} - S_{ch})(e|Z_a|/k_B T)(\delta L/L)$, where $\delta L$ is the length of single-gated region and $L$ is the full channel length. Together with Mott's formula for Seebeck

coefficient $S = (\pi^2 k_B^2 T/3e)\text{dln}\,\sigma/\text{d}E_F \sim (\pi^2 k_B^2 T/3eE_F)$ this yields $R_{TE}$ of the order of 5–20 V/W for Fermi energies in the range 50–200 meV which is more than two orders of magnitude smaller than the measured responsivity of our TFET detector yet close to that of conventional FET detectors based on gapless monolayer graphene[52]. Thus, the variations of the "bulk" thermoelectric parameters cannot explain the observed strong and asymmetric photoresponse of our device.

We note, however, that it is rather challenging to test whether the tunnel junction itself acts as a thermoelectric generator or not[56]. The respective thermoelectric coefficient can be estimated as $S_{TJ} \sim (k_B^2 T/e)\text{dln}\,G_{TJ}/\text{d}V_g$, and its functional dependence on the gate voltage would be indistinguishable from the junction rectification described by our model. In principle, measurements of the electron temperature can elucidate the presence of such rectification mechanism, which is however beyond the scope of our work. Nevertheless, even if present, such a mechanism is also due to the presence of the tunnel junction that substantiates the use of TFETs for sensitive THz detection.

Last but not least, we note that reaching the ultimately-low-noise-equivalent power would require impedance matching between antenna and TFET[57]. As the noise power density is proportional to $r_{2pt}$, and the voltage responsivity saturates at large resistances[58], we can expect the reduction of noise level by a factor of $[r_{2pt}/Z_{rad}]^{1/2}$ in optimized devices. Taking $Z_{rad} = 75 \, \Omega$ and $r_{2pt} = 1$–5 kΩ at the points of maximum $R_v$, we anticipate the ultimate NEP to be 3–8 times smaller than that reported in Fig. 3c and Supplementary Fig. 3. The simplest way of such matching lies in increasing the device width $W$.

In conclusion, we have shown an opportunity to use TFETs as high-responsivity detectors of sub-THz and THz radiation. Constructing a prototypical device from a BLG dual-gated structure and coupling it to a broadband antenna allowed us to demonstrate the drastic difference between a conventional FET-based approach and TFET-enabled rectification. Furthermore, we have developed a full model enabling one to predict the performance of TFET detectors based on the details of their band structure. This model was applied to the case of BLG-TFET detector and successfully captured all the experimentally observed features. As an outlook, we note that BLG is just a convenient

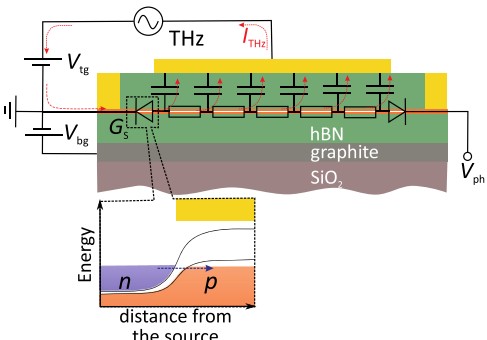

**Fig. 5 Equivalent circuit of the BLG-TFET detector.** Antenna is modelled as an equivalent voltage source $V_{ant}$ that generates ac current $I_{THz}$ (red arrows) flowing into the source and escaping the FET channel through the gate capacitance. Rectification occurs mainly at the tunnel barrier between source and channel (see band alignment profile in the inset) with voltage-dependent conductance $G_S$. The doping level and the band gap size is controlled via a simultaneous action of the top and bottom gate voltages, $V_{tg}$ and $V_{tg}$, respectively. The photovoltage, $V_{ph}$, is read out between the source and drain terminals.

platform to demonstrate the performance of TFET-based THz detectors. This approach can be extended to larger-gap materials[59] enabling room-temperature operation, as well as to CMOS-compatible structures[60]. Furthermore, we envision that alternative transistor technologies enabling transconductance beyond Boltzmann limit (phase-change FETs[61], negative capacitance FETs[62]) would also demonstrate ultra-sensitive THz detection.

## Methods

**Device fabrication.** To fabricate tunnelling-enabled BLG photodetector we first encapsulated BLG between relatively thick hBN crystals using the standard dry-peel technique[49]. The thickness of hBN crystals was measured by atomic force microscopy. The stack was then transferred on top of a predefined back gate electrode made of graphite deposited onto a low-conductivity THz-transparent silicon wafer capped with a thin oxide layer (500 nm). The resulting van der Waals heterostructure was patterned using electron beam lithography to define contact regions. Reactive ion etching was then used to selectively remove the areas unprotected by a lithographic mask, resulting in trenches for depositing electrical leads. Metal contacts to BLG were made by evaporating 3 nm of chromium and 60 nm of gold. Afterwards, a second round of e-beam lithography was used to design the top gate. The graphene channel was finally defined by a third round of e-beam lithography, followed by reactive ion etching using poly(methyl methacrylate) and gold top gate as the etching mask. Finally, a fourth round of e-beam lithography was used to pattern large bow-tie antenna connected to the source and the top-gate terminals, followed by evaporation of 3 nm of Cr and 400 nm of Au. Antennas were designed to operate at an experimentally relevant frequency range.

**Responsivity measurements.** To perform the photoresponse measurements we used variable temperature optical cryostat equipped with a polyethylene window that allowed us to couple the photodetector to incident THz radiation. A Zytex-106 infrared filter was mounted in the radiation shield of the cryostat to block the 300 K background radiation. The radiation was focused to the bow-tie antenna by a silicon hemispherical lens attached to the silicon side of the chip. The transparency of the silicon wafers to the incident THz radiation over the entire $T$-range was verified by transmission measurements using a THz spectrometer. Photovoltage was recorded using a home-made data acquisition system based on the PXI-e 6363 DAQ board.

The responsivity of our tunnelling-enabled detector was calculated assuming that the full power delivered to the device antenna funnelled into the FET channel. The $R_v$ determined by this way provides a lower bound for the detectors' responsivity and is usually referred to as extrinsic. The calculation procedure comprised several steps. First the drain-to-source voltage was recorded as a function of the top gate voltage in the dark ($V_{dark}$). Then, the dependence of the the drain-to-source voltage, $V_{DS}$, on $V_{tg}$ was obtained under the illumination with THz radiation. The latter was provided by a calibrated backward wave oscillator generating $= 0.13$ THz radiation with the output power $P_{out} \approx 1$ mW accurately measured using a Golay cell. To ensure the characterization of the detector in the linear-response regime, $P_{out}$ was further attenuated down to $P_{full} \approx 2$ μW, being the

full power delivered to the cryostat window. The difference $V_{ph} = V_{DS} - V_{dark}$ formed the photovoltage. The responsivity was then calculated as $R_v = V_{ph}/P_{in}$, where $P_{in} \approx P_{full}/3.5$ is the power delivered to the antenna after taking into account the losses in the silicon lens and the cryostat optical window ($\approx 5.5$ dB).

In order to study the photoresponse of our detectors at higher $f$, we used a quantum cascade continuous wave laser based on a GaAs/Al$_{0.1}$Ga$_{0.9}$As heterostructure emitting $f = 2.026$ THz radiation. Due to the low power of the QCL and non-optimized antenna design at this $f$, the calibration of the delivered to the device antenna power was rather challenging and therefore we only report tunnelling-enabled operation of our detector in relative units.

**Rectification modelling.** Our detector can be modelled by an equivalent circuit (shown in Fig. 5) comprising an effective voltage source $V_{ant}$ mimicking an antenna and two nonlinear junctions connected in series with transistor channel. The detector asymmetry, required to obtain a finite photovoltage at zero bias, is provided by the asymmetric connection of antenna between source and gate, and by zero-current condition at the drain. Calculation of detector voltage responsivity $R_v = V_{ph}/P_{in}$ includes three distinct steps:

- Relating the nonlinear $I(V)$ characteristics of circuit elements to the rectified voltage $V_{ph}$.
- Relating the power incident on antenna with its open-circuit voltage $V_{ant}$.
- Microscopic calculation of $I(V)$ characteristics for BLG channel and its tunnel contacts.

First, it is convenient to introduce "voltage-voltage" responsivity of the TFET, $R_{TFET} = V_{out}/V_{ant}^2$. The responsivity of bare transistor channel coupled to antenna between source and drain is the log-derivative of the dc channel conductance $G_{ch}$ with respect to the top-gate voltage $V_{tg}$[21], up to a geometrical factor:

$$R_{ch,i} = -\frac{1}{2} \frac{d_b}{d_t + d_b} \frac{\partial \ln G_{C,dc}}{\partial V_{tg}}. \tag{2}$$

The presence of a tunnel junction with conductance $G_S$ (assumed frequency-independent) depending on the voltage at the junction $V_{TJ}$ and the top-gate voltage $V_{tg}$ results in two extra contributions to $R_{TFET}$, which also take the form of log-derivatives:

$$R_{TJ,i} = -\frac{1}{2} \frac{\partial \ln G_S}{\partial V_{TJ}}, \qquad R_{TG,i} = -\frac{\partial \ln G_S}{\partial V_{tg}}. \tag{3}$$

Summation law (1) for individual responsivities (2) and (3) follows directly from Kirchhoff's circuit rules.

At the second stage, the experimentally measured "voltage-power" responsivity of the photodetector $R_v$ is related to the "voltage-voltage" responsivity of the TFET as

$$R_v = 4Z_{rad} \left| \frac{Z_{GS}}{Z_{GS} + Z_{rad}} \right|^2 R_{TFET}, \tag{4}$$

assuming the incident radiation is focused within the antenna's effective aperture. The prefactor describes voltage division between the TFET impedance $Z_{GS}$ between gate and source and antenna radiation resistance $Z_{rad}$.

Finally, we calculate the $I(V_d, V_{tg})$-characteristics of circuit elements microscopically. The FET channel is described within drift-diffusive model with constant mobility $\mu_{BLG} = 10^5$ cm$^2$/(V s), a value close to that found in the experiment. Both junctions are described within quantum ballistic model[54]. Both the flux of carriers incident on tunnel barrier and its transparency depend on BLG band structure. This model results in an approximate relation for source junction conductance $G_S \approx \frac{2e}{\pi^{3/2}\hbar} \mathcal{D}_{tun} k_{\perp tun} W$, where $\mathcal{D}_{tun}$ is the barrier transparency for normal incidence and $k_{\perp tun}$ is the characteristic transverse momentum of electrons participating in tunnelling. To obtain vanishing junction resistance in the absence of tunnel barrier, $\mathcal{D}_{tun}$ was replaced by $\mathcal{D}_{tun}/(1 - \mathcal{D}_{tun})$[63]. The appearance of high-transparency regions across the tunnel barrier due to local electric potential fluctuations was modelled as an increase of the average field inside the tunnel barrier $F_{tun}$ by a constant value $F_{fluct}$. A value of $F_{fluct} \approx 8$ kV/cm was extracted from the experimental resistance $R_{2pt}$ in the tunnelling regime of detector operation.

The calculation of TFET band structure in the double-gated and single-gated regions is based on a parallel-plate capacitor model supplemented with relations between charge densities on graphene layers, their electric potentials, and BLG band structure[64]. The transient region with tunnel junction was modelled using an original approach, where screening by the charges in BLG was treated approximately by placing a fictitious conducting plane under BLG. The position and potential of this plane are chosen to yield the correct electric potential deep inside the source and channel regions of the BLG. This reduces our electrostatic problem to finding the fringing field of a capacitor, solved analytically by Maxwell[65].

## Data availability

All data supporting this study and its findings are available within the article and its Supplementary Information or from the corresponding authors upon reasonable request.

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

## Acknowledgements

This work was supported by the Russian Foundation for Basic Research within Grants No. 18-37-20058 and No. 18-29-20116 and by the Ministry of Science and Higher Education of the Russian Federation (No. 0714-2020-0002). Experimental work of I.G. (photoresponse measurements) was supported by the Russian Foundation for Basic Research (grant 19-32-80028). We acknowledge support of the Russian Science Foundation grant No. 19-72-10156 (NEP analyses) and grant No. 17-72-30036 (transport measurements). The work of G.A. and D.S. (theory of THz detection) was supported by grant # 16-19-10557 of the Russian Scientific Foundation. K.W. and T.T. acknowledge support from the Elemental Strategy Initiative conducted by the MEXT, Japan, Grant Number JPMXP0112101001, JSPS KAKENHI Grant Number JP20H00354 and the CREST(JPMJCR15F3), JST. D.A.B. acknowledges financial support from Leverhulme Trust. The authors thank A. Lisauskas, W. Knap, A. I. Berdyugin, Q. Ma, and M.S. Shur for helpful discussions.

## Author contributions

D.S. and D.A.B. conceived the experiment. S.G.X. fabricated devices designed by D.A.B. Photoresponse measurements were carried out by I.G., M.M., and D.A.B. Data analysis was performed by I.G., D.A.B., and D.S. Theory analysis was done by G.A. and D.S. The manuscript was written by D.A.B., G.A., and D.S. with input from I.G. and G.F. Experimental support was provided by I.T., M.M., G.G., and A.K.G.; T.T. and K.W. grew the hBN crystals. D.S., G.F., and D.A.B supervised the project. All authors contributed to discussions.

## Competing interests

The authors declare no competing interests.
