## [Peer Review File · Nature Communications]

Reviewers' Comments:

Reviewer #1:

Remarks to the Author:

In their manuscript, Gayduchenko et al. present a new mechanism for THz detection that enables a strong improvement over standard field-effect transistors based on graphene. A responsivity beyond 3kV/cm was found at cryogenic temperature, which is more than one magnitude larger than the one for similar devices. The key to this improvement is a lateral tunneling barrier that is stemming from a gap opening in the bilayer graphene when top and bottom gate voltage is applied. These findings go significantly beyond earlier publications and justify publication in Nature Communications.

The manuscript is well written, the figures are clear and helpful to understand the working principle of the device. Nevertheless, one point is not clear to me and should be probably revised: in the sketches that provide the band profiles in Fig. 4 and the band structure in Fig. 1b, there is a clear band gap in both areas, the dual-gated and the single gated part at the edge of the channel. On the other hand, the dual gating is necessary for forming of the bandgap in the bilayer graphene (?). Thus, I wouldn't expect a gap opening in the single gated area next to the contacts! This should be explained and/or justified in more detail.

Two minor points:

- in Fig. 4 it might be helpful to also draw the band profile for the negative back gate voltage (i.e. -1.5V bg and 6V tg).
- related to that, I find the supplementary figure S1 very interesting and would like to encourage the authors to include it into the main manuscript!

Reviewer #2:

Remarks to the Author:

In this manuscript, the authors report on substantial enhancement of sensitivity of field-effect-transistor based detector which employs tunnel current injection (TFET). It is quite formidable that TFETs, although being extremely promising for digital electronics, were considered to possess low cut-off frequencies and their applicability in the terahertz frequency range was questionable. Here, the authors show, that for a graphene-based TFET such limitation might be lifted - this is one of the most interesting and novel aspect which, in my opinion, qualify the manuscript to be recommended for publication.

Therefore, it would be quite helpful if the authors could elucidate this aspect in more detail by addressing following questions: what are anticipated cut-off frequencies (in most relevant definition - as for detector) for their device, and how it comes that the capacitance of the tunnel junction is not shunting tunnel current. The theory presented in supplementary part seems to omit such mechanism (following equation S5), please give arguments why it is applicable for devices operating in THz frequency range.

Continuing with high-frequency characteristics, it would be very important to know how the impedance of device depends on frequency for different bias conditions. The knowledge of the impedance (total and of sub circuits like gated part and tunnel junction) is detrimental for efficient power transfer. Is it true, that right now the theory assumes impedance matching to antenna at every bias point, which is slightly unphysical.

In the literature, there are a lot of reports showing that in graphene-based devices a heating of transition between gated and ungated parts play a strong role to the amplitude and sign of detected signal. The so-called Seebeck contribution was considered as most plausible mechanism which can oppose mixing signals under the gate and even exceed the "standard" mechanism of rectification. Would it be possible to assess how far the tunnelling contribution exceed Seebeck mechanism?

Finally, I would like to advise to revise the references on the state-of-the-art as presented in Fig.

S3. For example, recent reports on liquid nitrogen cooled FETs are about three orders of magnitude better, the information on Schottky diodes could include devices from their leading producers (Toptica GmbH is just a distributor) and belong to the lower bound, cryogenically cooled Schottky devices improve their performance as well. Furthermore, backward-tunnel diodes are well known to be free from "Boltzmann tyranny" and for frequencies near 100 GHz are reported to have the similar NEP as reported here, but even at room temperature. Nevertheless, I consider that the concept of utilizing graphene as a detector in TFET configuration is novel and unique, bringing advantages of distributed channel in form of weakened impedance roll-off, thus are worth being published.

Reviewer #3:

Remarks to the Author:

The submitted work focuses on THz detection using FET like structures enhanced by a bi-layer graphene for improved sensitivity and noise equivalent power. The authors show impressive results, which are supported by considerations of BLG.

The following remarks on the text should be clarified before publication:

1. In the introduction the authors rightly state that the transconductance has a major impact on detector sensitivity. However, later in the text there seems to be a mixture of FET related standard transconductance limitation due to thermally excited carriers and the modulation of the channel resistance at sub-threshold and low drain voltages. Especially, the motivation arguments remain unclear and the statement "In spite of this variety, the use of TFETs for the rectification of high-frequency signals [29] has not been attempted so far." Is very misleading, as tunneling based rectification is an old phenomenon, which has been widely used in the past, but which was inefficient at that time. It would help to present the TFET as a way to increase the nonlinearity of the channel modulation with RF signal.

2. From Fig. 1e it becomes clear that the device is rather strongly mismatched with the antenna. According to numerous papers by several groups, including e.g. the group in Frankfurt and Warsaw, it has been demonstrated that antenna mismatch plays a vital role in sensitivity and NEP. Could the authors please comment on the mismatch losses.

Could the authors please also explain the asymmetry in the resistance curves in Fig. 1e. It seems there is a "rest" resistance towards lower voltages and the origin of this remains unclear.

3. The gate length of the TFET is rather large ($> 2\mu\text{m}$) compared to standard technologies. What is the impact of shorter gate lengths and why was the gate length chosen that long

4. The ungated regions of the TFET are rather short compared to the gate length. The authors do not indicate the reason for that. Is it to decrease the ungated resistances, which are not modulated by the RF. If so, how is it ensured that the THz signal is not impinging on the ungated regions.

5. Could the authors please indicate how the source and drain contacts have been realized and how these contact the graphene and hBN regions. It is especially interesting to see how the contact with hBN layers is avoided, which is crucial for the electrically-induced band gap in the BLG.

6. The results presented in figure 2 should be explained in more detail: the normalized results in Fig. 2d and Fig. 2e relate the channel resistance r_{2pt} and R_v . This assumes that the responsivity is directly related to the channel resistance in the gated and ungated regions. However, if a thermoelectric effect is present or carrier pushout would be present, similar results would be obtained. Could the authors please comment on this. It is not clear to the reviewer why the voltage responsivity is provided and not the more natural current responsivity (which is only deduced from voltage responsivity).

7. The evaluation of the NEP provided here is misleading. Given the formula $NEP=S/R_v$ means that the more input power is impinging on the detector the higher is the NEP. This is a little counter intuitive. Could the authors please clarify.

8. The operation of the BLG-TFET is not fully clear. As indicated in Fig. 1 and Fig. 5 one needs two electrodes in order to achieve the results. However, it is not clear, why a drain-source voltage would NOT create the required bandgap in the b-doped channel. Of the BLG structure.

9. Responsivity measurements: the amount of output power at 130 GHz and 2 THz are missing. Could the authors please indicate these values with verification procedures. It would also be interesting to learn how the standing-wave problem has been avoided in the measurements, especially due to the several junctions (vacuum window, lens, etc) and what is the spot size of the radiation on the sample.

10. Fig. 5 illustrates an equivalent circuit of the BLG-TFET. In this equivalent circuit the intrinsic channel is modelled by a distributed circuit of resistors shunted with capacitors. This is a rather traditional and accepted approach. The ungated regions are modelled with two diodes, representing the tunneling current flow, in series with the channel resistors. This would imply that the equivalent noise voltages of these add. Given the channel resistor of a value of around 200 Ohm it is not clear how the authors can arrive at such low noise equivalent power levels.

11. Reference 11 lacks year

Reviewer #1 (Remarks to the Author):

In their manuscript, Gayduchenko et al. present a new mechanism for THz detection that enables a strong improvement over standard field-effect transistors based on graphene. A responsivity beyond 3kV/cm was found at cryogenic temperature, which is more than one magnitude larger than the one for similar devices. The key to this improvement is a lateral tunneling barrier that is stemming from a gap opening in the bilayer graphene when top and bottom gate voltage is applied. These findings go significantly beyond earlier publications and justify publication in Nature Communications.

We thank the Reviewer for the careful reading of our manuscript and recommending the publication of our work in Nature Communications.

The manuscript is well written, the figures are clear and helpful to understand the working principle of the device.

We thank the Reviewer for this high assessment.

Nevertheless, one point is not clear to me and should be probably revised: in the sketches that provide the band profiles in Fig. 4 and the band structure in Fig. 1b, there is a clear band gap in both areas, the dual-gated and the single gated part at the edge of the channel. On the other hand, the dual gating is necessary for forming of the bandgap in the bilayer graphene (?). Thus, I wouldn't expect a gap opening in the single gated area next to the contacts! This should be explained and/or justified in more detail.

The band gap in bilayer graphene originates from the difference in on-site energies between the top and bottom graphene layers and depends on the average displacement field applied to the device [McCann and Vladimir I. Fal'ko, PRL 96, 086805 (2006), Y. Zhang, Nature 459, 820–823(2009)]. Indeed, the dual-gate configuration, pointed out by the Reviewer, is the desired geometry which enables one to reach *large* band gap values. However, even in the single-gated configuration, the energy difference is sufficient to open up a finite band gap as schematically illustrated in our Fig. 1b [see e.g. Castro et al., PRL 99, 216802 (2007)]. Following the Reviewer's suggestions, we have amended the draft to include this remark (see caption to Fig. 1b).

Two minor points:

- in Fig. 4 it might be helpful to also draw the band profile for the negative back gate voltage (i.e. -1.5V bg and 6V tg).

We thank the Reviewer for this suggestion. We have included the band profile for the case of negative V_{bg} and positive V_{tg} as the inset to Fig. 2a.

- related to that, I find the supplementary figure S1 very interesting and would like to encourage the authors to include it into the main manuscript!

In the revised draft, Fig. 2a was amended to address the Reviewer's recommendation: namely, we included the data for negative V_{bg} from Fig. S1.

Reviewer #2 (Remarks to the Author):

In this manuscript, the authors report on substantial enhancement of sensitivity of field-effect-transistor based detector which employs tunnel current injection (TFET). It is quite formidable that TFETs, although being extremely promising for digital electronics, were considered to possess low cut-off frequencies and their applicability in the terahertz frequency range was questionable. Here, the authors show, that for a graphene-based TFET such limitation might be lifted - this is one of the most interesting and novel aspect which, in my opinion, qualify the manuscript to be recommended for publication.

We thank the Reviewer for the careful reading of our manuscript, finding our results interesting and novel, and recommending our manuscript to be published.

Therefore, it would be quite helpful if the authors could elucidate this aspect in more detail by addressing following questions: what are anticipated cut-off frequencies (in most relevant definition - as for detector) for their device, and how it comes that the capacitance of the tunnel junction is not shunting tunnel current. The theory presented in supplementary part seems to omit such mechanism (following equation S5), please give arguments why it is applicable for devices operating in THz frequency range.

Indeed, shunting of the rectifying elements by their own capacitances can be an important obstacle for THz rectifiers. However, a detailed analysis shows that this problem does not arise in our device at least up to 5 THz which is the anticipated cut-off frequency. Such a cut-off appears when the junction capacitance becomes comparable to the antenna impedance ($\sim 50 \dots 100 \text{ Ohm}$), which is certainly not the case of our BLG detector at the frequencies studied in this work.

Let us explain this issue in more detail. The simplest equivalent circuit of an antenna-coupled rectifier that accounts for its own capacitance is shown in Fig. R1 [taken from A. Sanchez, C.F. Davis, K.C. Liu, and A. Javan, J. Appl. Phys. 49, 5270 (1978)]. The rectified current of the diode is $\frac{1}{4} (d^2I_d/dV^2) V_d^2$, where the amplitude of ac signal reaching the diode is $V_d = V R_d || Z_c / (R_a + R_d || Z_c)$. Here, V is the antenna open-circuit voltage, Z_c is the capacitive impedance of the diode, and $||$ stands for parallel connection of resistances. From above, we see that the diode voltage would drop below the antenna voltage if only $R_d || Z_c \ll R_a$. This is not the case of our BLG detector, because both R_d and Z_c have resistances of the order of kOhm . Indeed, the

capacitance of a two-dimensional junction is (up small logarithmic terms) $C = 2 W \epsilon \epsilon_0$, (see Eq. (13) in S.G. Petrosyan and A. Ya Shik. "Contact phenomena in low-dimensional electron systems." *Sov. Phys. JETP* 69 p. 1261 (1989)), where W is the device width and ϵ is hBN dielectric constant. The associated impedance is $|Z_c| = 1/(2 \pi f C) \sim 3.5 \text{ k}\Omega$ at $f=100 \text{ GHz}$ substantiating our modeling approach. When the radiation frequency is raised to $\sim 4\text{-}5 \text{ THz}$, the capacitive impedance becomes comparable to the radiative resistance of the antenna $\sim 75\text{-}100 \text{ }\Omega$. This is the anticipated cutoff range.

Fig. R1. The simplest equivalent circuit of an antenna-coupled THz rectifier. Taken from A. Sanchez, C.F. Davis, K.C. Liu, and A. Javan, *J. Appl. Phys.* 49, 5270 (1978)].

We have added the respective discussion to the last section of the amended manuscript:
"An important advantage of TFET rectifiers with 2d channels is the low internal capacitance of lateral tunnel junctions....."

Last but not least, we would like to mention two conclusions from the above arguments. First, if one is interested in the maximization of the voltage responsivity, the matching between antenna and rectifier is not critical. The voltage responsivity saturates to a maximum value if the device resistance is well above the antenna impedance. This agrees with the conclusions of [A. Sanchez, C.F. Davis, K.C. Liu, and A. Javan, *J. Appl. Phys.* 49, 5270 (1978), section IV A "Rectification"]. Despite being counter-intuitive, this is true as far as the photovoltage is proportional to the square of the applied voltage amplitude. For other mechanisms of rectification, where the photovoltage is proportional to dissipated power (e.g. thermal mechanisms), impedance matching is of course necessary to ensure maximum responsivity. Second, if one is interested in the lowest NEP, the matching is necessary again, because the noise increases with increasing the device resistance. In our device, NEP has not reached an optimal value due to relatively high tunnel resistance. However, since the thermal noise is proportional to the square root of the device resistance, the increase in noise due to such a

mismatch is only $\sim\sqrt{1\text{...}3 \text{ k}\Omega/100 \text{ }\Omega} = 3\text{...}5.5$ times. These estimates emphasize that the performance of the TFET detectors, proposed in our work, can be even further improved.

Continuing with high-frequency characteristics, it would be very important to know how the impedance of device depends on frequency for different bias conditions. The knowledge of the impedance (total and of sub circuits like gated part and tunnel junction) is detrimental for efficient power transfer. Is it true, that right now the theory assumes impedance matching to antenna at every bias point, which is slightly unphysical.

Our theory does not assume a perfect power transfer and perfect matching between the antenna and the device. The theory assumes that the antenna acts as a perfect voltage source, which is justified as far as the antenna impedance is well below the device impedance. The typical antenna impedance is of the order of 100 Ω , the device impedance is several $\text{k}\Omega$, therefore, in terms of the power transfer, the antenna and rectifier are unmatched. As we mentioned in the reply to the previous question, matching is unnecessary for voltage responsivity maximization, as soon as the rectified signal is proportional to the square of the applied voltage amplitude. From the point of view of NEP optimization, matching is an important requirement, because large resistance of the tunnel junction increases the noise level. However, since the noise spectral density (root mean square noise voltage) is proportional to the square root of the device resistance, the increase in noise in our device due to a mismatch is only $\sim\sqrt{1\text{...}3 \text{ k}\Omega/100 \text{ }\Omega} = 3\text{...}5.5$ times. The NEP at the best matching conditions can be several times smaller than that reported in our work, which further substantiates the promise of TFET-enabled THz detectors.

We have added a comment on this in the end of the discussion section of the revised manuscript:

“Last but not least, we note that reaching the ultimately-low noise-equivalent power would require impedance matching between antenna and TFET. ...”

In the literature, there are a lot of reports showing that in graphene-based devices a heating of transition between gated and ungated parts play a strong role to the amplitude and sign of detected signal. The so-called Seebeck contribution was considered as most plausible mechanism which can oppose mixing signals under the gate and even exceed the "standard" mechanism of rectification.

We agree with the Reviewer that the thermoelectric effect at the interface between regions of different doping can also contribute to the rectification of high-frequency radiation in graphene FETs - this indeed has been proven to be an important mechanism as discussed in numerous papers including ours [*Nat. Comm.* **9**, 5392 (2018) and *Applied Physics Letters* **112**, 141101 (2018)]. However, first, we would like to notice that in the coupling geometry schematically illustrated in Fig. 1a-d (antenna is connected between the source and gate terminals whereas

the signal is measured between the source and drain electrodes), the thermoelectric rectification, standard resistive self-mixing, as well as the demonstrated in this work tunneling-assisted rectification have the same sign of the emerged photovoltage. Indeed, an interplay between the former two mechanisms was extensively studied by our group in APL **112**, 141101 (2018) which provided qualitative and quantitative evidence that there is in fact no competition between thermoelectric and resistive self-mixing photoresponse but instead they work in consonance. In particular, both effects result in *positive* photovoltage when the gated-region is p-doped and non-gated (or single-gated in this case) region is n-doped and in negative photovoltage in the opposite case provided that the antenna-coupling configuration is such as that shown in Fig. 1a-d. As it follows from our data and calculations (e.g. Fig. 2a) the sign of the photovoltage due to the tunneling-enabled rectification follows an identical trend excluding competition with previously known mechanisms.

Second, experimentally, on a V_{tg}/V_{bg} map, the conventional thermoelectric rectification would reveal itself via a pronounced six-fold photovoltage pattern as demonstrated in several experiments on graphene in the frequency range from microwaves and THz to visible light (see e.g. Science **334**, 648-652 2011, Nano Lett. **19**, 2765–2773 (2019), Nano Lett. **16**, 6988–6993 (2016)). This stems from the fact that the thermoelectric photovoltage is proportional to the difference in Seebeck coefficients of the single- and dual-gated regions, which in turn, according to the Mott's relation, are proportional to the derivative of the conductivity with respect to the Fermi energy. Such a difference results in the double change of sign in the photovoltage dependence on one of the gates when the voltage on another gate is fixed. Our experimental map in Fig. 3a is characterized by a clear four-fold pattern indicating that the thermoelectric contribution is rather weak if present.

Would it be possible to assess how far the tunnelling contribution exceed Seebeck mechanism?

The theory for the Seebeck effect in antenna-coupled FETs was developed in our earlier paper Applied Physics Letters **112**, 141101 (2018). According to these results, the emerging responsivity is

$$R = \frac{3}{2\pi^2} [S_{ch} - S_{cont}] \frac{eZ_A}{k_B T} \frac{\delta L}{L}$$

Here, S_{ch} and S_{cont} are the Seebeck coefficients in the graphene channel, and in the near-contact regions, respectively, δL is the length of the near-contact region and L is the full channel length, Z_A is the radiative resistance of the antenna. In the present manuscript, the “near-contact region” is the single-gated part of the BLG. The equation was derived by solving the heat conduction equation for electrons and substituting the distribution of temperature $T(x)$ into a formula for the thermoelectric voltage $V = - \int [S(x) dT(x)]$. For the step-like profile of the

Seebeck coefficient, the expression is simplified greatly. The schematic of heating power density, Seebeck coefficient, and temperature are shown in Fig. R2. Next, using the Mott's relation for the Seebeck coefficient of degenerate electron system, we get

$$S = \frac{\pi^2}{3} \frac{k_B^2 T}{e} \frac{d \ln \sigma}{d E_F} \sim \frac{\pi^2}{3} \frac{k_B^2 T}{e E_F}$$

Taking $\delta L = 300$ nm, $L = 6$ μ m, $Z_A = 100$ Ohm, $T = 10$ K, $E_F = 50 \dots 200$ meV, we obtain $R = 6 \dots 20$ V/W, which is more than two orders of magnitude smaller than the measured responsivity of our TFET detector yet close to that of conventional FET detectors based on gapless monolayer graphene.

Fig. R2. Distribution of the Seebeck coefficient S , dissipated heat q , and temperature T in a FET with the radiation coupled between the source and gate terminals. Taken from APL **112**, 141101 (2018).

This low value is not unusual: the junction where the thermoelectric voltage is developed is located close to the cold contact. We can speculate more generally that THz detectors with the Dyakonov-Shur coupling scheme (i.e. when the antenna is connected between the source and gate terminals) are not much suited for thermoelectric detection. The thermoelectric effect is most pronounced in devices where the junction is located in the 'hot spot' of the field (see e.g. S. Castilla et.al. "Fast and sensitive terahertz detection using an antenna-integrated graphene pn junction" *Nano letters*, 19(5), 2765-2773 (2019))

Let us briefly mention the assumptions used in this model and their validity:

- (1) The thermal conduction equation for electrons is solved by neglecting the energy loss into the phonon bath. The energy loss is associated with the diffusion of energy into cold contacts only. Inclusion of cooling by phonon emission is possible but it would further reduce the responsivity due to the thermoelectric effect.

- (2) The ac current heating the electron system is assumed to flow only over the left half of the channel. In reality, it is distributed over all the channel and its magnitude is gradually reduced from source to drain. The inclusion of a more realistic current distribution would also reduce the estimate of responsivity, as the distribution of temperature would become more symmetric (asymmetry of heat distribution is the necessary condition for the emergence of photovoltage).

As a sub-conclusion, we can state that thermoelectric voltage emerging due to the variation of the *bulk* Seebeck coefficient is negligible in our situation.

- (3) The last assumption of our model is the continuity of the temperature at the p-n junction itself. This means that the junction does not have its own thermal resistivity. Related to that, we have assumed that thermoelectric voltage emerges due to the jump in *bulk* Seebeck coefficient. These assumptions are widely adopted in the literature [Q. Ma, N.M. Gabor, T.I. Andersen, N.L. Nair, K. Watanabe, T. Taniguchi, and P. Jarillo-Herrero, Phys. Rev. Lett. 112, 1 (2014); J.C.W. Song, M.S. Rudner, C.M. Marcus, and L.S. Levitov, Nano Lett. 11, 4688 (2011)] yet require better justification. For example, they are justified when the thermal conduction of the p-n (tunnel) junction is well above the thermal conduction of the bulk channel. As the heat and charge conductivities are bound with Wiedemann-Franz law, the junction would not introduce thermal resistance as soon as its electrical resistance is low. Strictly speaking, it is not the case in the tunneling regime. We can suggest that a tunnel junction can block the heat leakage into a cold contact, and introduce strong electron overheating. The thermoelectric effect at the tunnel junction itself is also possible and is the order of

$$S_{tun} \sim \frac{k_B^2 T}{e^2} \frac{d \ln G_{tun}}{dV}$$

Experimentally, it is rather challenging to distinguish between direct rectification and the Seebeck effect at the tunnel junction using electrical measurements only. In both cases, the photovoltage is proportional to log-derivatives of the tunnel conduction with respect to the voltage drop at the junction. The distinction is possible only with measurement of electron temperature (e.g. with noise thermometry).

Anyway, heat leakage blocking and the emergence of the thermoelectric voltage at the tunnel junction are the tunneling-induced phenomena. Therefore, our main idea about tunneling being the prerequisite of sensitive THz detection remains valid.

We thank the Referee for raising this question. Stimulated by this inquiry, we have added the discussion of respective phenomena in a new “Discussion” section of the revised manuscript which starts from:

“The measured photoresponse has been successfully described assuming direct rectification of THz signal by tunnel junction nonlinearity. ...”

Finally, I would like to advise to revise the references on the state-of-the-art as presented in Fig. S3. For example, recent reports on liquid nitrogen cooled FETs are about three orders of magnitude better, the information on Schottky diodes could include devices from their leading producers (Toptica GmbH is just a distributor) and belong to the lower bound, cryogenically cooled Schottky devices improve their performance as well. Furthermore, backward-tunnel diodes are well known to be free from "Boltzmann tyranny" and for frequencies near 100 GHz are reported to have the similar NEP as reported here, but even at room temperature.

We thank the Reviewer for this suggestion. We have amended the reference list related to Fig. S3 and included extra information on cooled FETs ($1\text{pW}/(\text{Hz})^{0.5}$ at $T=77\text{ K}$, Ref. S18), cooled Schottky devices (No NEP is reported at 77K but Refs. S19-S21 are included) and room-temperature backward-tunnel diodes ($2\text{pW}/(\text{Hz})^{0.5}$ at $T=300\text{ K}$, Ref. S22, and S23).

Nevertheless, I consider that the concept of utilizing graphene as a detector in TFET configuration is novel and unique, bringing advantages of distributed channel in form of weakened impedance roll-off, thus are worth being published.

We thank the Reviewer for finding our approach novel and unique and recommending it for a publication in Nature Communications.

Reviewer #3 (Remarks to the Author):

The submitted work focuses on THz detection using FET like structures enhanced by a bi-layer graphene for improved sensitivity and noise equivalent power. The authors show impressive results, which are supported by considerations of BLG.

We thank the Reviewer for carefully reading our manuscript and finding our results impressive.

The following remarks on the text should be clarified before publication:

1. In the introduction the authors rightly state that the transconductance has a major impact on detector sensitivity. However, later in the text there seems to be a mixture of FET related standard transconductance limitation due to thermally excited carriers and the modulation of the channel resistance at sub-threshold and low drain voltages. Especially, the motivation arguments remain unclear and the statement “In spite of this variety, the use of TFETs for the rectification of high-frequency signals [29] has not been attempted so far.” Is very misleading, as tunneling based rectification is an old phenomenon, which has been widely used in the past, but which was inefficient at that time. It would help to present the TFET as a way to increase the nonlinearity of the channel modulation with RF signal.

Indeed, the use of tunneling-enabled rectifiers has been reported in the past. We have acknowledged these results in the following sentence in the main text: *“This is also surprising considering recent advances in the development of tunneling high-frequency rectifiers and detectors based on quantum dots (30,31), diodes (32-37) and superconducting tunnel junctions(38-40)”*. However, to the best of our knowledge, it is the TFET configuration that has not been used before for this inquiry. We have managed to find only one reference on the subject and purely of theoretical consideration (Ref. 29). This indeed allowed us to conclude that *“In spite of this variety, the use of TFETs for the rectification of high-frequency signals [29] has not been attempted so far.”* In the manuscript, we also emphasized the advantage of transistor-based detectors with respect to more conventional diode rectifiers (see the first paragraph of the main text) whereas the rest of the paper does present the results in the context of nonlinearity enhancement in accord with the Reviewer’s inquiry.

2. From Fig. 1e it becomes clear that the device is rather strongly mismatched with the antenna. According to numerous papers by several groups, including e.g. the group in Frankfurt and Warsaw, it has been demonstrated that antenna mismatch plays a vital role in sensitivity and NEP. Could the authors please comment on the mismatch losses.

Our BLG device is indeed mismatched with the antenna (device resistance is several kOhm, while typical antenna impedance is the order of 100 Ohm). This mismatch was fully accounted for in our measurements and modeling. When measuring the voltage responsivity $R = V_{ph}/P$, we used the net power P reaching the device. We did not make any recalculations of the incident power into absorbed one, and, therefore, only reported the *external* responsivity. In the modeling, we treated the antenna as a voltage source with internal resistance Z_A . This model is similar to the one used by ‘Frankfurt group’ in [M. Bauer, A. Ramer, S.A. Chevtchenko, K.Y. Osipov, D. Cibiraite, S. Pralgauskaite, K. Ikamas, A. Lisauskas, W. Heinrich, V. Krozer, and H.G. Roskos, IEEE Trans. Terahertz Sci. Technol. 9, 430 (2019)], except for the fact that ungated regions in our case have very strong nonlinearity due to tunneling. The power mismatch in this model is seen automatically as far as $Z_{FET} \gg Z_A$.

Importantly, the antenna-transistor matching is not necessary if the goal is to maximize the voltage responsivity. Indeed, $V_{ph} \sim (d \ln G/dV) V_{ac}^2$, where V_{ac} is the ac voltage drop at the rectifying element. When the transistor impedance is large, V_{ac} saturates to antenna open-circuit voltage, and voltage responsivity is maximized. On the other hand, in some applications, the maximization of the voltage responsivity is not as important as the minimization of the NEP (maximization of SNR). To achieve that, Z_A should be of the order of Z_{FET} (precise calculations in Supplementary section V show that it occurs at $Z_{FET} \approx 3Z_a$). As the thermal noise spectral density is proportional to the square root of the device resistance, the increase in noise level due to a mismatch is only $\sim \sqrt{1...3 \text{ kOhm}/100 \text{ Ohm}} = 3...5.5$ times in our BLG

device. The noise at best-matching conditions can be several times smaller than that reported here, which further substantiates the promise of TFET-enabled THz detectors.

We have added the respective discussion to the last section of the amended manuscript:

“Last but not least, we note that reaching the ultimately low noise-equivalent power would require impedance matching between antenna and TFET...”

Could the authors please also explain the asymmetry in the resistance curves in Fig. 1e. It seems there is a “rest” resistance towards lower voltages and the origin of this remains unclear.

Indeed, there is a strong asymmetry of the resistance curves with respect to the top gate voltage when a finite V_{bg} is applied. Importantly, as we commented in the main text, this “rest” resistivity has an insulating temperature dependence (Fig. 3d inset) and appears when the band alignment profile is such as shown in Fig. 1a, i.e. when the dual- and single-gated regions are separated by a tunnel junction. This allows us to conclude that this rest resistivity originates from the tunnel junction itself. Note, when $V_{bg}=0$ V the $r_{2pt}(V_{tg})$ dependence is rather symmetric (Fig. 1e inset). In the revised draft we have added a sentence pointing out to this asymmetry:

“However, r_{2pt} exhibits clear asymmetry with respect to the CNP...”

3. The gate length of the TFET is rather large ($> 2\mu\text{m}$) compared to standard technologies. What is the impact of shorter gate lengths and why was the gate length chosen that long.

The length of the top gate electrode was chosen to be of the order of plasmon wavelength at the frequencies of available to us radiation sources. Our initial hope was to enhance the photoresponse via plasmon resonance as demonstrated in our previous work (*Nat. Comm.* **9**, 5392 (2018)). Despite the pronounced plasmon resonances in our device, no amplification was observed (See Fig. S2). However, as we discuss in the main text, the dominant rectification stems from the *localized* tunnel junction but not from the BLG channel on either side of the junction. Therefore, we would not expect a drastic difference if the top gate electrode was shorter.

4. The ungated regions of the TFET are rather short compared to the gate length. The authors do not indicate the reason for that. Is it to decrease the ungated resistances, which are not modulated by the RF. If so, how is it ensured that the THz signal is not impinging on the ungated regions.

The reason for a relatively short length of the single-gated region was the same as that discussed in the answer to the previous question. In particular, we aimed to maximize the dual-gated region where we anticipated to reach high-quality resonance of acoustic (gated) plasmons [*Nat. Comm.* **9**, 5392 (2018)] and thereby enhance the photoresponse. The length of the single-gated region would not affect the resulting photoresponse because the dominant contribution occurs at the tunnel junction (not in the single or double-gated channels).

5. Could the authors please indicate how the source and drain contacts have been realized and how these contact the graphene and hBN regions. It is especially interesting to see how the contact with hBN layers is avoided, which is crucial for the electrically-induced band gap in the BLG.

An exact schematic on how the source and drain metal leads contact the bilayer graphene is provided in Fig. 1a of the manuscript. To achieve it, we removed the top hBN layer in designated areas using a highly selective reactive ion etching recipe which, after hBN removal, leaves bilayer graphene intact as described in Methods. After this, we deposited chromium (3nm)/gold (60 nm) leads using standard e-beam evaporation (a more detailed description of the fabrication of such contacts can be found in Nat. Phys. **12**, 318–322 (2016)). The photograph of the resulting device is shown in Fig. 1c. We do not avoid contact with hBN layers: our top gate electrode is deposited on top of the top hBN layer. hBN is an insulating material and therefore acts as a gate dielectric.

6. The results presented in figure 2 should be explained in more detail: the normalized results in Fig. 2d and Fig. 2e relate the channel resistance r_{2pt} and R_v . This assumes that the responsivity is directly related to the channel resistance in the gated and ungated regions. However, if a thermoelectric effect is present or carrier pushout would be present, similar results would be obtained. Could the authors please comment on this. It is not clear to the reviewer why the voltage responsivity is provided and not the more natural current responsivity (which is only deduced from voltage responsivity).

Photoresponse studies of THz detectors can be performed by measuring either the buildup photovoltage or the photocurrent that emerged in response to incident radiation. The choice usually depends on the internal resistance of the detector as compared to the load/measurement circuitry. For example, if the detector resistance is large compared to the input impedance of the voltmeter, i.e. as in the case of FETs in the OFF-state, the apparent voltage responsivity is not going to reflect the detector performance as accurately demonstrated in Sakowicz et al., J. Appl. Phys. 110, 054512 (2011). On the contrary, if the detector resistance is small compared to the load resistance, the photocurrent configuration is also not going to provide quantitative characteristics of the detector's rectification ability as it will depend on the load resistance. Thus, one should choose the best measurement configuration for a specific detector type to probe the physical mechanisms governing the photoresponse.

The resistance of our tunneling-enabled detector varies between $r_{2pt}=0.5-20$ kOhm depending on the V_{tg}/V_{bg} combination. This allows one to choose either of the measurement methods. Indeed, the input impedance of our voltage measurement board is >10 GOhm that is much larger than r_{2pt} , whereas, for a typical operation of the detector as a current source, a 50 Ohm

load is used that is much smaller than r_{2pt} . Because our photovoltage measurements do not involve the use of additional load resistors connected in series, they provide the intrinsic performance of the TFET-based THz detector and fully justify our choice. By relating R_v to r_{2pt} in Figs. 2d-e, we simply plot the part of the current responsivity which would not depend on the load resistance of the circuitry.

7. The evaluation of the NEP provided here is misleading. Given the formula $NEP=S/R_v$ means that the more input power is impinging on the detector the higher is the NEP. This is a little counter intuitive. Could the authors please clarify.

The evaluation of the NEP using the protocol described in our draft is a rather standard approach applied since the earliest works on graphene-based THz detectors [see e.g. Nat. Mat. **11**, 865–871 (2012), APL **104**, 061111 (2014), Carbon **116**, 760-765 (2017), IEEE TRANS. ON THZ SCIE. AND TECHN. **7**, N. 5, (2017) and others]. More input power does not change the result because the responsivity R_v , defined as a ratio between the photovoltage and the radiation power, is a power-independent quantity in our experiments (Fig. R3). This indicates that the detector operates in the linear regime, in which the electronic temperature remains comparable to that of the lattice. In addition, we did not observe any bolometric effects, i.e. device resistance remained unaffected by the power of incident radiation. These observations advocate for the power-independent NEP of our device and fully justify our approach.

Fig. R3. R_v as a function of V_{tg} recorded in response to 0.13 THz radiation for a different power of the incoming radiation. $P_0=0.5 \mu\text{W}$.

8. The operation of the BLG-TFET is not fully clear. As indicated in Fig. 1 and Fig. 5 one needs two electrodes in order to achieve the results. However, it is not clear, why a drain-source voltage would NOT create the required bandgap in the b-doped channel. Of the BLG structure.

As shown in Fig. 4b, depending on the chosen $V_{\text{tg}}/V_{\text{bg}}$ configuration we can control the band alignment profile between the dual- and single-gated regions. For some $V_{\text{tg}}/V_{\text{bg}}$ combinations, one can achieve the band alignment profiles, such as those shown in the insets to revised Fig. 2a, which resemble that of typical TFETs. The gap in the band structure of BLG originates from the *perpendicular* electric field applied across graphene layers which we apply using the top and bottom gate electrodes. The in-plane electric field from the source-drain voltage does not open the band gap in the BLG band structure (and we do not apply any source-drain voltage in our experiments).

9. Responsivity measurements: the amount of output power at 130 GHz and 2 THz are missing. Could the authors please indicate these values with verification procedures. It would also be interesting to learn how the standing-wave problem has been avoided in the measurements, especially due to the several junctions (vacuum window, lens, etc) and what is the spot size of the radiation on the sample.

We thank the reviewer for this inquiry. In the revised draft we have provided the output power of our calibrated BWO source which was of the order of 1 mW. This was accurately measured using a standard Golay cell. To ensure the characterization of the detector in the linear-response regime, P_{out} was further attenuated down to $P_{\text{full}} \approx 2 \mu\text{W}$, being the full power delivered to the cryostat window. We have also checked the transparency of our cryostat windows and silicon lens at this frequency and revealed losses of the order of 5.5 dB as indicated in the Methods section of our manuscript. This was done by means of standard transmission experiments. These losses were accounted for when calculating the responsivity of our tunnel-enabled detector (see Methods). In addition, we have checked the transparency of the substrate, on which the detector is assembled, to the 0.13 THz radiation using a home-build spectrometer. As per 2 THz radiation, we used a helium-cooled GaAs-based quantum cascade laser. However, it was rather challenging to determine the delivered power due to the unstable operation of the laser, difficulties in measuring the transmission of the substrate, and cryostat window. For this reason, we avoid quantitative statements of our detector's performance at this frequency and only demonstrated that the responsivity dependence on the gate voltage preserves its strong asymmetry indicating tunneling-enabled photoresponse as discussed in the main text. Note, we have amended the text and provided the anticipated cut-off frequency of our TFET-based detector which was found to be well above the frequency range studied in this work.

The spot size of the radiation on the sample is determined by the hemispherical silicon lens to which our detector is attached. Because our device is in the focus of the lens, the spot size is given by the dimensions of the Airy disk of the incident radiation (the antenna size is also comparable to the size of the Airy disk). To minimize the reduction of the delivered power as a

result of the standing wave formation, the position of the cryostat with respect to the radiation source was optimized to provide the strongest possible photovoltage.

10. Fig. 5 illustrates an equivalent circuit of the BLG-TFET. In this equivalent circuit the intrinsic channel is modelled by a distributed circuit of resistors shunted with capacitors. This is a rather traditional and accepted approach. The ungated regions are modelled with two diodes, representing the tunneling current flow, in series with the channel resistors. This would imply that the equivalent noise voltages of these add. Given the channel resistor of a value of around 200 Ohm it is not clear how the authors can arrive at such low noise equivalent power levels.

The NEP levels which we present in the manuscript are the results of two experiments: 1) responsivity and 2) resistivity measurements. The peak responsivity in our devices reaches 3 kV/W whereas the resistivity of our device at this V_{tg}/V_{bg} combination is of the order of 1 kOhm. Given that the temperature at which the detector operates is 10 K, one obtains the noise spectral density of the order of $S=(4k_B r_{2pt} T)^{0.5} = 7.4 \times 10^{-10} \text{ V/Hz}^{0.5}$. The resulting NEP is the ratio between S and R_v which for this V_{tg}/V_{bg} combination is as low as 0.2 pW/sqrt(Hz) as reported in Fig. 3c.

11. Reference 11 lacks year

We thank the Reviewer for noticing this. We have amended the reference list to account for this inquiry.

Reviewers' Comments:

Reviewer #1:

Remarks to the Author:

The authors well respondet to all points raised, I recommend publication of the current mansucript!

Reviewer #2:

Remarks to the Author:

The authors gave thorough explanations to all my questions and performed appropriate modifications of the manuscript. I would like to recommend publishing this article as is.

Reviewer #3:

Remarks to the Author:

The reviewer thanks the authors for their replies and explanations. This clarifies many aspects of the paper. It seems that the question 10 of reviewer 3 has not been answered properly. The response refers to NEP discussion while the question relates to equivalent circuit.

Reviewer #1 (Remarks to the Author):

The authors well respondet to all points raised, I recommend publication of the current mansucript!

We thank the Reviewer for recommending the publication of our manuscript.

Reviewer #2 (Remarks to the Author):

The authors gave thorough explanations to all my questions and performed appropriate modifications of the manuscript. I would like to recommend publishing this article as is.

We thank the Reviewer for careful reading of our manuscript and for recommending its publication.

Reviewer #3 (Remarks to the Author):

The reviewer thanks the authors for their replies and explanations. This clarifies many aspects of the paper.

We are grateful to the Reviewer for careful reading of the revised version of our manuscript.

It seems that the question 10 of reviewer 3 has not been answered properly. The response refers to NEP discussion while the question relates to equivalent circuit.

In fact, the Reviewer's question #10 from does refer to the NEP:

*10. Fig. 5 illustrates an equivalent circuit of the BLG-TFET. In this equivalent circuit the intrinsic channel is modelled by a distributed circuit of resistors shunted with capacitors. This is a rather traditional and accepted approach. The ungated regions are modelled with two diodes, representing the tunneling current flow, in series with the channel resistors. This would imply that the equivalent noise voltages of these add. **Given the channel resistor of a value of around 200 Ohm it is not clear how the authors can arrive at such low noise equivalent power levels.***

We fully agree with the Referee's description of our equivalent circuit, and with the fact that the noise spectral power densities (mean square voltages) of the channel and the tunnel diodes are additive. However, we stress that the reported low values of NEP ($0.2 \text{ pW/Hz}^{0.5}$) are not derived by any means from the equivalent circuit. The reported NEP represents the ratio of the noise spectral density $S=(4k_B r_{2pt} T)^{0.5}$, evaluated with the *experimentally* measured resistance r_{2pt} of the device, and the *experimentally* determined responsivity R_v . Importantly, r_{2pt} is the full two-terminal resistance measured between the source and drain terminals and it accounts for the resistance of the dual-gated region (channel), tunnel junction resistance, resistance of the single-gate part of BLG as well as the contact resistance between the BLG and gold leads. When calculating the NEP of our device, we use r_{2pt} because, in accord with the Reviewer's comment, it depends on both channel and the tunnel junction resistance and therefore it is responsible for the total noise level. We emphasize that the details of the equivalent circuit do not affect the reported NEP being a combination of the *experimentally measured* characteristics. The direct evaluation of the NEP with such measured r_{2pt} leads to $S=7.4 \times 10^{-10} \text{ V/Hz}^{0.5}$ (at $T=10 \text{ K}$, $r_{2pt} \sim 1 \text{ kOhm}$). Dividing that value by responsivity $R_v=3 \text{ kV/W}$, we obtain NEP as low as $0.2 \text{ pW/Hz}^{0.5}$ reported in the main text.